# Development of a Subunit Vaccine against Duck Hepatitis A Virus Serotype 3

**DOI:** 10.3390/vaccines10040523

**Published:** 2022-03-28

**Authors:** Trang-Nhu Truong, Li-Ting Cheng

**Affiliations:** 1International Degree Program of Animal Vaccine Technology, International College, National Pingtung University of Science and Technology, 1, Shuefu Road, Neipu, Pingtung 91201, Taiwan; truongtrangnhu@gmail.com; 2Institute of Veterinary Research and Development in Central Vietnam, Km 4, Road 2/4, Vinh Hoa, Nha Trang City 57000, Vietnam; 3Graduate Institute of Animal Vaccine Technology, College of Veterinary Medicine, National Pingtung University of Science and Technology, 1, Shuefu Road, Neipu, Pingtung 91201, Taiwan

**Keywords:** Duck hepatitis A virus serotype 3, VP1, flagellin, subunit vaccine, adjuvant

## Abstract

In this study, we sought to develop a subunit vaccine against the increasingly prevalent Duck hepatitis A virus serotype 3 (DHAV-3). The VP1 protein of DHAV-3 and a truncated version containing the C-terminal region of VP1, termed VP1-C, were expressed recombinantly in *Escherichia coli* as vaccine antigens. For enhanced immune response, a truncated version of flagellin, *n*FliC, was included as vaccine adjuvant. Ducklings were vaccinated once for immune response analysis and challenge test. Results showed that VP1-C elicited a higher level of virus-specific antibody response and neutralization titer than VP1. The addition of *n*FliC further enhanced the antibody response. In terms of cellular immune response, the VP1-C + *n*FliC vaccine elicited the highest level of T cell proliferation among the vaccine formulations tested. Examination of the cytokine expression profile showed that peripheral blood mononuclear cells from the VP1-C + *n*FliC vaccine group expressed the highest levels of pro-inflammatory (IL-6) and TH-1 type (IL-12 and IFN-γ) cytokines. Finally, in a DHAV-3 challenge test, the VP1-C + *n*FliC vaccine provided a 75% protection rate (*n* = 8), in contrast to 25% for the VP1 vaccine. In conclusion, *E. coli*-expressed VP1-C has been shown to be a promising antigen when combined with *n*FliC and may be further developed as a single-dose subunit vaccine against DHAV-3.

## 1. Introduction

Duck hepatitis A virus (DHAV) infection of ducklings results in liver necrosis and ecchymotic hemorrhage, with mortality rates reaching 95% [1,2,3]. There are currently three antigenically unrelated genotypes in circulation, DHAV-1, DHAV-2, and DHAV-3 [4,5,6,7]. DHAV-1 is the most prevalent worldwide and DHAV-2 is limited to Taiwan [1,3,8,9].

DHAV-3 has been reported in Korea, China, and recently Vietnam [1,3,9]. Since DHAV-3 is becoming increasing prevalent in Vietnam, we aimed to develop a safe and efficacious subunit vaccine against DHAV-3.

As the only member of the *Avihepatovirus* genus of the *Picornaviridae* family, DHAV is a nonenveloped virus with a single-stranded, positive-sense RNA genome [10,11,12,13]. The genome encodes a long open reading frame that is translated into a precursor polyprotein [10,14]. While being translated, the precursor polyprotein is also processed by nascent viral proteinases [15,16,17]. The polyprotein can be divided into three regions: P1, P2, and P3 [12,17,18]. The P1 region encodes the capsid proteins, while the P2 and P3 regions encode proteins for genome replication and protein processing [11,12,18,19]. For virus capsid assembly, the P1 precursor protein folds nascently and is cleaved into capsid proteins VP0, VP3, and VP1, forming the first intermediate structural unit, the protomer [10,19,20,21]. Five protomers then assemble into a pentamer and finally 14 pentamers pack together to form an icosahedral capsid [22,23,24,25]. The three viral capsid proteins do not have sequence homology, but nonetheless, all fold similarly into a wedge-shaped, eight-stranded β-barrel [19,20,26,27]. The eight antiparallel β-strands form two β-sheets that comprise the wedge structure whereas the connecting loops and the C-terminus are surface-exposed and contain important neutralizing antigenic sites [16,18,19,20,26]. The N-terminus, on the other hand, resides on the inside of the virion and forms a network of protein–protein interactions important for pentamer formation and capsid stability [18,20].

In a previous study, the VP1 protein of DHAV-1 has been shown to elicit neutralizing antibodies that may interfere with receptor-binding [1]. We therefore aimed to evaluate the protective efficacy of DHAV-3 VP1 in ducklings. In addition to expressing the full VP1 recombinantly in *Escherichia coli* as the antigen, we also expressed a shorter recombinant protein containing only the C-terminal portion (residues 149 to 218) of VP1, termed VP1-C. The shorter VP1-C may allow better protein-folding and also contains the antigenically important, surface-exposed C-terminus. Furthermore, since purified protein antigens usually do not elicit a strong immune response, we evaluated the benefit of adding a biological adjuvant, flagellin. Flagellin is a pathogen-associated molecular pattern that activates Toll-like receptor 5, leading to inflammatory immune response and adaptive immunity [28,29]. In our previous work, the N-terminus of flagellin, termed *n*FliC, was shown to be sufficient for adjuvancy [28]. Therefore, *n*FliC was employed as an adjuvant in this study.

Overall, this study aimed to evaluate the protective efficacy of DHAV-3 VP1 and VP1-C when adjuvanted with *n*FliC. Ducklings were immunized once with formulated vaccines for immune response analysis and challenge test.

## 2. Materials and Methods

### 2.1. DHAV-3 Virus Culture

A field strain of DHAV, named NT01 (OK631673), was isolated from a duck farm in Ninh Thuan province and confirmed to be DHAV-3 using primers for the VP1 gene (Table 1). Briefly, liver samples were collected from ducklings showing opisthotonus posture and homogenized by grinding through a mesh in 0.9% saline. Samples were filtered through 0.2 µm Millex-FG syringe filters (Merk KGaA, Darmstadt, Germany) and the filtrates were stored at −80 °C as stock samples. The virus was propagated successfully, with visible cytopathic effect (Appendix A), in the Leghorn Male Hepatoma cell line (ATCC ^®^ CRL-2118TM) at 37 °C, 5% CO_2_, in Waymouth’s Medium (Gibco Invitrogen, Carlsbad, CA, USA) supplemented with 10% FBS (Fetal Bovine Serum, Gibco Invitrogen, Carlsbad, CA, USA). Virus presence in the cell culture (Appendix A) was reconfirmed with reverse transcription quantitative PCR using primers qVP1-F and qVP1-R (Table 1) designed based on the VP1 gene, indirect immunofluorescence antibody (IFA), as well as independent challenge tests (Appendix A). TCID_50_ of the cultured virus was determined by end-point dilution assay and reached 10^4.68^.

For VP1 gene cloning and sequencing, total RNA extraction of the cultured virus was performed with the Total RNA Extraction Miniprep System (Viogen, Taipei, Taiwan), followed by reverse transcription PCR using the High-Capacity cDNA Reverse Transcriptase Kit (Applied Biosystems, Foster, CA, USA) with VP1 primers listed in Table 1. The resulting PCR product was digested with the restriction enzymes EcoRI and HindIII, ligated into the plasmid vector pET32a (Novagen, Darmstadt, Germany), and sent for sequencing.

### 2.2. Recombinant Protein Expression of VP1, VP1-C, and nFliC

VP1, VP1-C, and *n*FliC genes were cloned and inserted into pET32a for recombinant protein expression in *E. coli*. Full-length VP1 gene was obtained as described in Section 2.1. For VP1-C (residues 149 to 218), PCR was performed using the VP1 gene as the template along with a pair of primers, VP1-C F and VP1-C R (Table 1). For the cloning of *n*FliC, a pair of primers (Table 1) was employed for PCR, with full-length FliC as the template [2]. PCR products were inserted into pET32a, and the resulting plasmid constructs were propagated in DH5α (Yeastern Biotech, Taipei, Taiwan). Sequence of the inserted genes were reconfirmed.

For recombinant protein expression, BL21 (DE3) *E. coli* cells (Yeastern Biotech, Taipei, Taiwan) were transformed with plasmid constructs according to the manufacturer’s instructions. Protein expression was induced with 1-mM isopropyl-b-D-galactopyranoside (IPTG; Sigma, Darmstadt, Germany) at 37 °C for 4 h. Subsequently, cells were lysed in denaturing lysis buffer (6 M Urea, 300 mM KCl, 50 mM KH_2_PO_4_ and 5 mM Imidazole) and sonicated. Protein expression was verified with 12% sodium dodecyl sulfate polyacrylamide gel electrophoresis (SDS-PAGE) analysis, using BSA (KPL, Gaithersburg, MD, USA) standards for protein quantitation. Western blotting was also performed to reconfirm the identity of the recombinant proteins. 6× His tag antibody (GeneTex, Hsinchu, Taiwan) at 1:5000 dilution was used as the primary antibody, with goat anti-mouse IgG antibody (HRP) (GeneTex, Hsinchu, Taiwan) at 1:5000 dilution as the secondary antibody. Western Lightning PLUS (PerkinElmer, Waltham, MA, USA) was used for color development. Finally, recombinant proteins were purified through the His tag using Bio-scale Mini Profinity IMAC cartridges (1 mL) (Bio-Rad, Hercules, CA, USA) before dialyzing against diminishing concentrations of urea (3 M, 1 M, 0.5 M, and ddH_2_O) at 4 °C. Endotoxin levels of the purified proteins were confirmed to be less than 0.125 EU/mL with ToxinSensor^TM^ Chromogenic LAL Endotoxin Assay Kit (GenScript, Piscataway, NJ, USA).

### 2.3. Vaccine Preparation and Immunization of Ducklings

Five vaccine formulations were prepared: (1) VP1, (2) VP1 + *n*FliC, (3) VP1-C, (4) VP1-C + *n*FliC, and (5) PBS as the vehicle control. Recombinant proteins were emulsified with the water-in-oil adjuvant Summit-P101 (Country Best Biotech, Taipei, Taiwan) at a 40:60 (water:oil) ratio. Each vaccination dose was 0.5 mL, containing 80 µg of recombinant protein. The vaccine formulations of VP1/VP1-C + *n*FliC contain 40 µg of antigen and 40 µg of *n*FliC.

Day-old Tsiay ducklings from a regional farm in Khanh Hoa province were assigned to five groups of 11 (three ducklings for immune response analysis and eight for the challenge test) and kept at the Institute Veterinary of Research and Development in Central Vietnam.

Ducklings were vaccinated once intramuscularly with the five vaccine formulations on Day 0. For immune response analysis, whole blood was collected weekly from three ducklings per vaccine group. All animal trial procedures followed the Ethical Rules and Law of the Animal Ethics Committee, and the protocol (AEC-NLU-20190905) was approved by the Animal Ethic Committee of Nong Lam University of Ho Chi Minh City (NLU), Vietnam.

### 2.4. Analysis of Humoral Immune Response

Levels of virus-specific antibody of the vaccinated ducklings were determined. Whole blood samples were permitted to coagulated and then centrifuged at 700× *g* for 5 min for serum collection. Indirect enzyme-linked immunosorbent assay (ELISA) was carried out by coating plates with 100 µL of 100 TCID_50_ NT01 overnight at 4 °C. After washing and blocking, serum samples at 1:10,000 dilution (0.5% skim milk in PBS) were added as the primary antibody. Goat HRP-conjugated anti-duck IgG (Sigma, Carlsbad, CA, USA) at a 1:5000 dilution was used as the secondary antibody. The Peroxidase kit (KPL, Gaitherburg, MD, USA) was used for color development, and optical density was read at 450 nm on the Multiskan^TM^ FC microplate photometer (Thermo Fisher Scientific, Vantaa, Finland).

To determine serum neutralization titers, sera were heated at 56 °C for 30 min to inactivate the complement. Sera were diluted two-fold serially, mixed with 100 TCID_50_ of NT01 and incubated for 1 h at 37 °C before applying to LMH cells (5 × 10^4^ cell/well) in 96-well plates. The cells were observed for six days for the appearance of CPE to calculate serum neutralization titers using the Reed and Muench method [30].

### 2.5. Analysis of Cellular Immune Response

To assess the cellular immunity elicited by the vaccines, T cell proliferation assay was performed. Peripheral blood mononuclear cells (PBMCs) were collected from immunized ducklings. Briefly, whole blood was collected using BD Vacutainer™ EDTA Blood Collection Tubes (BD Biosciences, Franklin Lakes, NJ, USA) and layered onto equal volume of Ficoll-Paque (Amersham Biosciences, Piscataway, NJ, USA) for centrifugation at 525× *g* for 45 min. The buffy coat layer containing PBMCs was collected, and the cells were rinsed twice with PBS before resuspension at 2 × 10^6^ cells/mL in RPMI-1640 (Gibco Invitrogen, Carlsbad, CA, USA) supplemented with 10% fetal bovine serum (Gibco Invitrogen, Carlsbad, CA, USA). For antigen stimulation, prepared PBMCs (4 × 10^5^ cells/well in 96-well plates) were incubated with 10 µg/mL of purified recombinant VP1 for 48 h at 37 °C and 5% CO_2_. Concanavalin A (Thermo Fishier Scientific, Ward Hill, MA, USA) at 5 µg/mL was used as the positive control for cell stimulation, with cell-only as the negative control and medium-only as background. The EZcount^TM^ MTT Cell Assay Kit (Himedia, Dubai, India) was used then to measure the extents of cell proliferation and the results were read at 560 nm. Stimulation index (SI) = (OD of treatment − OD of background)/(OD of the negative control − OD of background).

### 2.6. Analysis of Cytokine mRNA Levels

To examine the cytokine expression profile of lymphocytes from vaccinated ducklings, isolated PBMCs (2 × 10^6^ cells/well in 24-well plates) were stimulated with 10 µg/mL of purified recombinant VP1 for 3 h at 37 °C and 5% CO_2_. Total RNA was then extracted with the Total RNA Extraction Miniprep System (Viogene, Taipei, Taiwan) and cDNA synthesis was carried out using the High-Capacity cDNA Reverse Transcriptase Kit (Applied Biosystems, Foster, CA, USA).

Real-time PCR was set up in the SmartCycler I (Cepheid, Sunnyvale, CA, USA) with primers (Table 1) for cytokines and the housekeeping gene glyceraldehyde-3-phosphate dehydrogenase (GAPDH), which was used as an internal control for gene expression [31,32]. Cytokine expression levels were expressed as N-fold increase or decrease relative to that of the PBS control group, as calculated by the 2^−ΔΔCT^ method [33], assuming amplification efficiency approaches 100%.

### 2.7. Duck Hepatitis A Virus Serotype 3 Challenge Test

On Day 14, vaccinated ducklings (*n* = 8) were challenged intramuscularly with 100 TCID_50_ of NT01. Ducklings were monitored for clinical signs of DHAV infection for humane sacrifice: ruffled feather, wing drooping, ataxia, anorexia, watery diarrhea (green manure, dehydration) and depression. Surviving ducklings were sacrificed at the end of the 14-day observation period. Animal experimental protocols (AEC-NLU Approval No. 20190905) were approved by the Animal Ethic Committee, Nong Lam University of Ho Chi Minh City (NLU), Vietnam.

### 2.8. Statistical Analysis

Statistical analysis was performed using the Graphpad Prism software version 8.4.2 (San Jose, CA, USA). After confirming that data points follow a normal distribution by the Shapiro-Wilk test, one-way analysis of variance (ANOVA) and Tukey’s post-hoc test were used for mean comparison for data from antibody response, cytokine mRNA levels and T cell proliferation. All data are expressed as mean ± SD (standard deviation). In all cases, *p* < 0.05 was considered to be statistically significant.

## 3. Results

### 3.1. Recombinant VP1 and VP1-C Were Formulated as Vaccines

For the development of a subunit vaccine against DHAV-3, the VP1 gene was cloned from a DHAV-3 field isolate and expressed in *E. coli*. Purified recombinant VP1, along with VP1-C and *n*FliC have expected molecular weights of 46, 28, and 31 kDa, respectively (pET32a vector inserts a 20-kDa tag), in SDS-PAGE analysis (Figure 1A). Identities of the proteins were reconfirmed in Western blot analysis using anti-His antibody (Figure 1B, see Appendix A for full gel photos). Expression quantity of the recombinant proteins were: 152 mg/L for VP1, 213 mg/L for VP1-C, and 814 mg/L for *n*FliC. The amino acid sequence of VP1 is shown in Figure 1C. Furthermore, amino acid sequence comparison showed that the VP1 gene is 97.5% and 97.92% similar with that of DHAV-3 strains from China (KP715494.1, YT1213, China) and Vietnam (KU860089.1, NC, Vietnam), respectively (Figure 1D). Five vaccine formulations were prepared: (1) VP1, (2) VP1 + *n*FliC, (3) VP1-C, (4) VP1-C + *n*FliC, and (5) PBS as the vehicle control. Ducklings were vaccinated once intramuscularly for immune response analysis and challenge test.

### 3.2. VP-1C + nFliC Elicited the Highest Level of Virus-Specific Antibody and Neutralization Titer

To evaluate the humoral immunity induced by the vaccines, DHAV-3-specific antibody levels of vaccinated ducklings were determined using indirect ELISA. Results showed that, for all of the vaccine groups, antibody levels peaked on Day 14 after vaccination and decreased thereafter (Figure 2A). Comparing between the vaccine groups on Day 14, the VP1-C group showed a higher antibody level than the full-length VP1, indicating that the C-terminus of VP1 may serve as a better antigen. Furthermore, the addition of *n*FliC enhanced antibody levels for both VP1 and VP1-C. For end-pint dilution tittering of antibody levels, see Appendix A. Overall, the VP-1C + *n*FliC vaccine elicited the highest level of virus-specific antibody.

Sera from vaccinated ducklings were also evaluated for neutralizing antibody titer against DHAV-3. Neutralizing antibody was detected as early as Day 7 after vaccination and reached the highest levels on Day 14 for the vaccine groups (Figure 2B). On Day 14, the VP1-C group showed a higher neutralizing titer than the VP1 group. Furthermore, the addition of *n*FliC enhanced neutralizing titer for both VP1 and VP1-C. These neutralizing antibody results correlate well with that of virus-specific antibody. In summary, analysis of humoral immune response showed that VP1-C may be a superior antigen compared to full-length VP1 and that *n*FliC can enhance antibody titers.

### 3.3. VP-1C + nFliC Elicited the Highest Level of T Cell Immune Response

An antigen-specific T cell proliferation experiment was performed to evaluate the cellular immunity elicited by the vaccines. Results showed that on Day 14 after vaccination, VP1-C induced a higher level of T cell proliferation than VP1 (Figure 3). The addition of *n*FliC provided a boost to both VP1 and VP1-C. Consistent with the result of humoral immunity, the VP1-C + *n*FliC vaccine also induced the highest level of cellular immunity among the various formulations.

### 3.4. VP1-C + nFliC Elicited the Highest Level of ProInflammatory Cytokine Expression

To obtain a more detailed profile of the immune response elicited by the vaccines, cytokine expression of PBMCs from immunized ducklings (Day 14) was analyzed. Overall, the VP1-C + *n*FliC vaccine group showed the highest cytokine expression levels compared to other groups (Figure 4). In terms of the level of the pro-inflammatory cytokine IL-6, a significant difference can be seen between the VP1 and VP1-C vaccines. For the T_H_-1 type cytokines, IL-12 and IFN-γ, VP1-C appeared to provide a stronger boost than VP1, and *n*FliC also contributed to enhanced cytokine expression.

### 3.5. VP1-C + nFliC Provided 75% Protection Rate in a Challenge Experiment versus 25% for VP1

For protective efficacy evaluation, immunized ducklings were challenged with 100 TCID_50_ of the DHAV-3 isolate NT01 two weeks after vaccination. Survival rate for the VP1 vaccine group was 25%, with the VP1-C group slightly higher at 37.5% (Figure 5). With the addition of *n*FliC in the vaccine formulation, survival rate was significantly enhanced for the VP1-C group, rising to 75%. Survival rate of the VP1 group, however, showed only slight improvement with the addition of *n*FliC. In summary, VP1-C enhanced by *n*FliC provided a promising level of protection against DHAV-3 challenge.

## 4. Discussion

Our study showed that a single-dose vaccination of ducklings with the VP1-C + *n*FliC vaccine formulation provided a promising level of protection against DHAV-3 challenge. Analysis of humoral and cellular immunity confirmed the immunogenicity of VP1-C and demonstrated that the inclusion of *n*FliC provided a significant boost to immune response for VP1-C.

The results obtained in our study are quite similar to those observed for *E. coli*-expressed VP1 of enterovirus 71 [34]. Zhang et. al. showed that full-length VP1 of EV71 failed to elicit neutralizing serum in mice. Neutralizing activity was only observed when mice were vaccinated with truncated C-terminal portions of VP1: VP1_202–297_ or VP1_202–248_, both of which reside toward the C-terminus of VP1. Other works on EV71 also identified positions 163–177 and 208–222 (near the C-terminus) as linear neutralizing epitopes [30,35,36]. The result from our study correlates well with the observation that a truncated VP1 containing the C-terminal region elicits better protective immunity than full-length VP1. It remains an interesting question, though, as to why full-length VP1 fails to elicit protective immunity since it contains all the neutralizing epitopes. A hypothesis could be that protein folding of *E. coli*-expressed VP1 may result in the masking of neutralizing epitopes.

While *E. coli*-expressed VP1 appears ineffective as a vaccine antigen, baculovirus-expressed VP1 seems to elicit better neutralizing immunity. In one study, rabbit sera raised against baculovirus-expressed VP1 of DHAV-1 neutralized virus infection in vitro and in vivo [1]. In another study, baculovirus-expressed virus-like particle composed of DHAV-1 VP0, VP3, and VP1 was found to induce strong humoral immune response and provided strong protection [37]. The baculovirus expression system may lend the superior immunogenicity through post-translational modifications.

Various protein expression systems confer different advantages in terms of vaccine efficacy and costs. By showing that *E. coli*-expressed VP1-C may serve as an effective antigen, we have demonstrated that the *E. coli* expression system remains a viable option for DHAV antigen production.

## 5. Conclusions

In conclusion, we have shown *E. coli*-expressed VP1-C to be a promising antigen when combined with *n*FliC, and it may be further developed as a single-dose subunit vaccine against DHAV-3.

## Figures and Tables

**Figure 1 vaccines-10-00523-f001:**
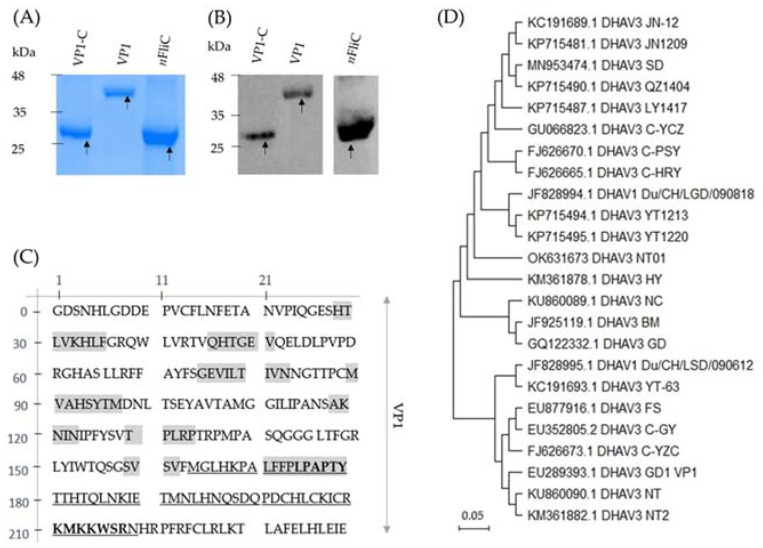
Recombinant protein expression of VP1, VP1-C, and *n*FliC. (**A**) SDS-PAGE and (**B**) Western blot analysis were performed for the purified recombinant proteins. (**C**) Protein sequence of cloned VP1 from the DHAV-3 strain NT01 used in this study is shown. Predicted β-strands (highlighted in grey), potential conserved B cell epitope, heparan-binding site (bold font), and VP1-C (underlined) are indicated. Prediction of β-strand locations is based on sequence comparison with the VP1 gene of the *Human Parechovirus 1* strain, Harris (protein data bank ID: 5MJV). (**D**) Phylogenetic analysis of NT01 VP1 amino acid was performed with 23 DHAV isolates of two serotypes (DHAV1 and DHAV3). Phylogenetic analysis was performed using the maximum likelihood method (in the MEGA 11 software).

**Figure 2 vaccines-10-00523-f002:**
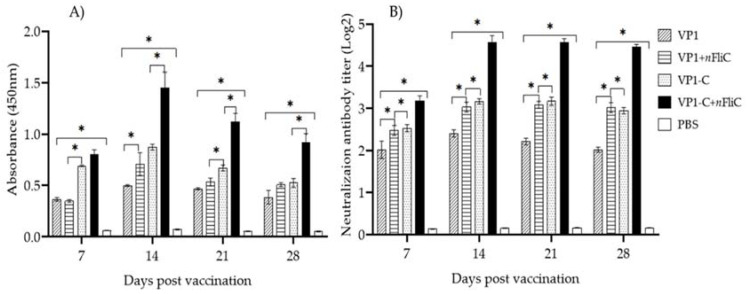
Virus-specific antibody levels of vaccinated ducklings. Ducklings (*n* = 3 per group) were vaccinated once with five different vaccine formulations: (1) VP1, (2) VP1 + *n*FliC, (3) VP1-C, (4) VP1-C + *n*FliC, and (5) PBS, and virus-specific antibody levels were determined by indirect ELISA. (**A**) IgG antibody level of vaccinated ducklings; (**B**) Neutralization antibody titer of vaccinated ducklings. Data are presented as mean ± SD. An asterisk indicates statistically significant difference (*p* < 0.05).

**Figure 3 vaccines-10-00523-f003:**
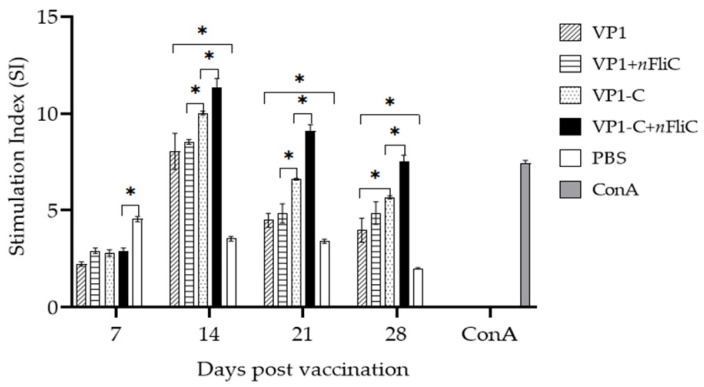
T cell proliferation of PBMCs from vaccinated ducklings. Ducklings (*n* = 3 per group) were vaccinated once with five different vaccine formulations: (1) VP1, (2) VP1 + *n*FliC, (3) VP1-C, (4) VP1-C + *n*FliC, and (5) PBS. Isolated PBMCs were stimulated with VP1, and proliferation extent was measured using the MTT assay. ConA was used as the positive control. Data are presented as mean ± SD. An asterisk indicates statistically significant difference (*p* < 0.05).

**Figure 4 vaccines-10-00523-f004:**
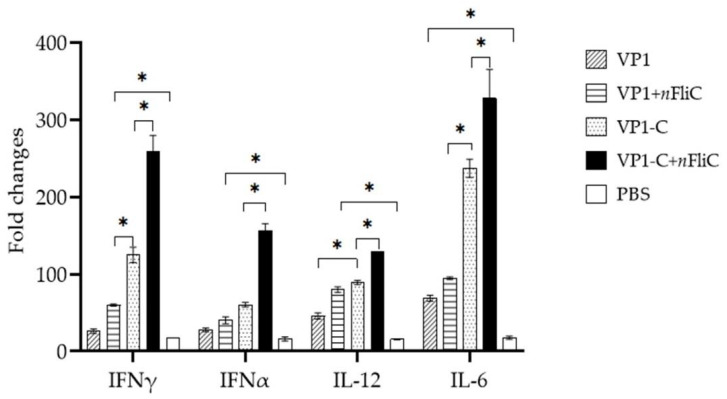
Cytokine expression profile of PBMCs from vaccinated ducklings. Ducklings (*n* = 3 per group) were vaccinated once with five different vaccine formulations: (1) VP1, (2) VP1 + *n*FliC, (3) VP1-C, (4) VP1-C + *n*FliC, and (5) PBS. Isolated PBMCs (Day 14) were stimulated with VP1, and RT-PCR was performed to determine mRNA levels of pro-inflammatory cytokines (IFNγ, IFNα, IL-12, and IL-6). Data are presented as mean ± SD. An asterisk indicates statistically significant difference (*p* < 0.05).

**Figure 5 vaccines-10-00523-f005:**
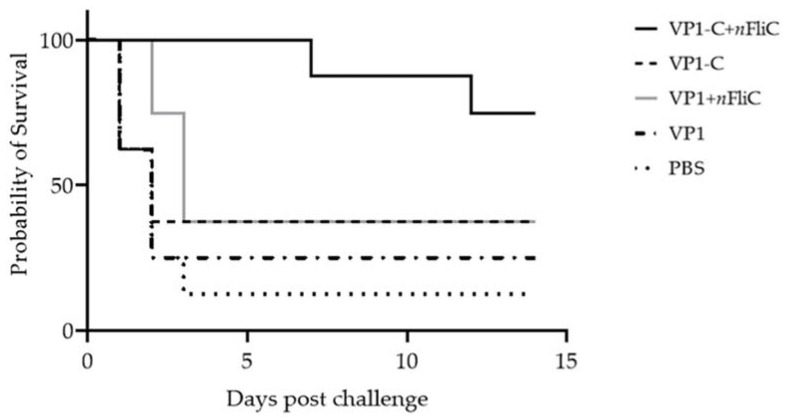
Survival rate of vaccinated ducklings after challenge with Duck hepatitis A serotype 3, NT01. Ducklings (*n* = 8 per group) were vaccinated once with five different vaccine formulations: (1) VP1, (2) VP1 + *n*FliC, (3) VP1-C, (4) VP1-C + *n*FliC, and (5) PBS. Two weeks after vaccination, ducklings were challenged with 100 TCID_50_ NT01. Survival rate was recorded for 14 days.

**Table 1 vaccines-10-00523-t001:** Primers for gene cloning and cytokine genes.

Target Gene		Sequence (5′-3′)	Gene-Length (bp)	Annealing Temp. (°C)	GenBank
VP1	F	**^1^ gaattc**atgggtgattctaatcatctt	732	53	OK631673
	R	**aagctt**ttcaatttccaaatggagct			
VP1-C	F	**gaatt**catgggcaggttgtatatctgg	222	53	OK631673
	R	**aagctt**attgcgagaccatttctt			
*n*FliC	F	**gaattc**atgaactgcactaaacaaact	309	52	[29]
	R	**aagctt**ttcagcctggatggagtc			
GAPDH	F	atcacaccacacatggcgt	207	59.5	GCA_015476345.1
	R	tttatagccgccgaggctg			
IFN-γ	F	cagacctactgcttgtttgt	192	53.5	AJ012254
	R	ttcatttctctctgtccagt			
IFN-α	F	cagccatctacagcgccc	138	60	AY879230.1
	R	ggctgggagccatgttgc			
IL-12	F	ctggaggtcattgatgaggtg	168	56	AJ564201
	R	gaaagtcaaagggaagtaggac			
IL-6	F	accgtgtgcgagaacagc	135	60	AB191038
	R	gaaaagcccgctggagagt			
DHAV-1	F	gccccactctatggaaatttg	767	53	[9]
	R	atttggtcagattcaatttcc			
DHAV-3	F	atgcgagttggtaaggattttcag	881	56	[9]
	R	gatcctgatttaccaacaaccat			
qVP1	F	atgaaccagtgtgttttctc	175	53	OK631673
	R	aacagagatgcatgaccc			

^1^ Bold font represents restriction enzyme (RE) sites. IFN-α, γ: interferon alpha, gamma; IL-6, 12: interleukin-6, 12; GAPDH: glyceraldehyde-3-phosphate dehydrogenase.

## Data Availability

The data displayed in this study is available on request from the corresponding author.

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
