# Peer review of "Development of a Subunit Vaccine against Duck Hepatitis A Virus Serotype 3"

_vaccines, 2022, doi:10.3390/vaccines10040523_

Round 1

Reviewer 1 Report

Truong TN and Cheng LT developed a subunit vaccine against Duck hepatitis A virus genotype 3 (DHAV-3) by expressing the truncated VP1 protein in Escherichia coli. The manuscript is much better after revision. However, the manuscript is still needed to be improved. Specific comments are listed below.

  1. In the supplementary figure S3, CPE was observed in LMH cells inoculated with DHAV-3. The authors need to provide immunostaining of viral antigen in LMH cells inoculated with DHAV-3.
  2. The authors stated that different superscript letters indicate significant differences (p < 0.05) between treatment groups at the same timepoint. I believed that the data is confusing. I suggested to present the data in a more straightforward way. For example, *, significant difference between VP1 group and PBS control. ^, significant difference between VP1-C group and PBS control. #, significant difference between VP1-C - and VP1+FliC groups. Please modify.

Author Response

In response to the comments:

1. In the supplementary figure S3, CPE was observed in LMH cells inoculated with DHAV-3. The authors need to provide immunostaining of viral antigen in LMH cells inoculated with DHAV-3.

Response: In the supplementary figure S3, we have added pictures of immunostaining of viral antigen in LMH cells inoculated with DHAV-3.

2. The authors stated that different superscript letters indicate significant differences (p < 0.05) between treatment groups at the same timepoint. I believed that the data is confusing. I suggested to present the data in a more straightforward way. For example, *, significant difference between VP1 group and PBS control. ^, significant difference between VP1-C group and PBS control. #, significant difference between VP1-C - and VP1+FliC groups. Please modify.

Response: We have revised our figures to use the asterisk for indicating significant differences (p < 0.05).

Reviewer 2 Report

I have carefully reviewed the new version of the article “Development of a subunit vaccine against Duck hepatitis A virus serotype 3 “ ( vaccines-1609252), and the response letter from the authors in which they indicate point by point the clarifications and modifications requested, checking that they have incorporated it into the manuscript.

I have also reviewed the suggestions of the other reviewers which have seemed very interesting and timely.

From my point of view, although this experience has limitations, it also allows us to advance in knowledge.

Author Response

We deeply appreciate the reviewer's help in improving our work.

Round 2

Reviewer 1 Report

I believed that the manuscript is much improved and will be ready for publication.

Author Response

Thank you for your support.

This manuscript is a resubmission of an earlier submission. The following is a list of the peer review reports and author responses from that submission.

Round 1

Reviewer 1 Report

Truong TN and Cheng LT developed a subunit vaccine for duck hepatitis A virus genotype 3 (DHAV-3) by employing truncated VP1 protein as antigen. High levels of antibody and T cell response were induced in ducklings after being administered with truncated VP1 protein. Of note, high protective rate was also observed in a DHAV-3 challenge test. Overall, the data are not well presented, and some of them seem unsound. There are several issues that the authors need to address.

  1. In “2.1. DHAV-3 Virus Culture” section, the authors reported that DHAV-3 could induce CPE in Leghorn Male Hepatoma cell line. Currently, no CPE was observed in DHAV-3 infected cells. Please provide more evidence to support their conclusion.
  2. Please combine table 1 and table 2.
  3. In table 2, the length of DHAV-1 VP1 gene is 714 bp rather than 720 bp. Please correct.
  4. In Figure 2, the antibodies level was determined by ELISA. Please provide the data about the ELISA titer but not absorbance at 450 nm. Neutralizing antibodies in serum should be also determined.
  5. In Figures 2, 3, and 4, the superscript letters were used in statistical analysis. It is not straightforward way to present the data. Please specify the meaning of individual letter in this study.
  6. The number of animals in DHAV-3 challenge assay is not limited. I was wondering if the authors validated the data by using independent assay, and please provide the data collected from independent experiment.

Author Response

November 19th, 2021

Vaccines

Dear Editor,

We are submitting a revised manuscript entitled: “Development of a subunit vaccine against Duck hepatitis A virus serotype 3” by Trang-Nhu Truong, and Li-Ting Cheng for your review for potential publication in Vaccines.

We are grateful for the reviewers’ supportive and helpful comments. Following is a table of point-by-point response to the comments.

Reviewer 2 Report

Thank you very much for allowing me to review the article entitled “Development of a subunit vaccine against Duck hepatitis A virus serotype 3” (vaccines-1455825).

The objective of this work is to develop a subunit vaccine against the increasingly prevalent Duck hepatitis A virus serotype 3 (DHAV-3). they conclude that E. coli-expressed VP1-C has been shown to be a promising antigen when combined with nFliC, and may be fur-ther developed as a single-dose subunit vaccine against DHAV-3.

This is a new article in the authors' line of research.

Introduction: It´s very well structured so that you can follow the fundamental guidelines on which the research is based, in a line of deepening in the subject. they are also adequately referenced. Suggestion: I suggest that at the end of the introduction the working hypothesis and objectives should be raised.

Material and methods: The virus culture in liver samples is described and the virus extraction from this sample was performed with the Total RNA Extraction Miniprep System (Viogen, Taipei, Taiwan), followed by reverse transcription PCR using the High-Capacity cDNA Reverse Transcriptase Kit (Applied BiosystemsTM, Foster, CA, USA) with VP1 primers. Subsequently recombinant Protein Expression of VP1, VP1-C, and nFliC and Vaccine Preparation and Immunization of ducklings. Finally, analysis of Humoral Immune Response, analysis of Cellular Immune Responses, analysis of Cytokine mRNA Levels, duck Hepatitis A Virus Serotype 3 Challenge Test and statistical analysis.

 In summary, the methodology is perfectly structured and explained, allowing us to understand each step that has been carried out.

 Results. The results are accompanied by tables and figures that allow a better understanding of the results obtained. it is very well structured.

Discussion: The discussion considered that it is the weak part of the work since it reflects on the result but does not consider that since it is another work in the line of the authors it could be compared with previous results and with published work on the topic could also raise possibilities for future research strengths and weaknesses. For all of which he considered that the discussion should be broadened.

Conclusions: the conclusion raised is consistent with the results and the objective of the study.

Author Response

November 19th, 2021

Vaccines

Dear Editor,

We are submitting a revised manuscript entitled: “Development of a subunit vaccine against Duck hepatitis A virus serotype 3” by Trang-Nhu Truong, and Li-Ting Cheng for your review for potential publication in Vaccines.

We are grateful for the reviewers’ supportive and helpful comments. Following is a table of point-by-point response to the comments.

Best regards,

Reviewer 3 Report

In this study (vaccines-1455825), in order to develop a safe and effective subunit vaccine against Duck hepatitis A virus (DHAV), which was renamed as Avihepatovirus A in 2014 by ICTV, the authors tried to demonstrate that recombinant bacterial proteins, which consist of thioredoxins (TrxA), polyhistidine-tag (6xHis-tag), S-tag and either full-length or C-terminal region of DHAV (serotype 3) VP1 protein, could induce antigen-specific host immune response and protect the vaccinated ducklings from lethal-dose challenge with pathogenic DHAV-3 strain. DHAV, a member of the family Picornaviridae, is a causative agent of the fatal duck viral hepatitis, which is ongoing threat to the duck industry. Nevertheless, currently, licensed (commercial) live-attenuated vaccines as well as inactivated vaccines are likely available for DHAV-1 but not other serotypes. As the VP1 protein, a major surface protein of pircornavirus particles, is the principal antigenic determinant, the protein should be a potential target for vaccine (and drug) development. Therefore, attempts of authors may contribute for the development of safe and low-cost vaccine against in the future. However, unfortunately, the quality of the manuscript is not enough to publish in Vaccines. 

-Major comments (There are too many. Only some of them are written below.)-
The reviewer surely agrees that the mixture of TrxA-6xHis-S-tagged C-terminal region (residues 149-218? see below comments) of VP1(VP1-C) and TrxA-6xHis-S-tagged N-terminus of flagellin (nFliC) as well as the water-in-oil adjuvant Summit-P101 could protect the injected duckling from lethal-dose of DHAV-3 challenge, somehow. However, it is hard to agree with the authors' main claim "vaccination with VP1-C and nFliC appeared to induce stronger host immune responses and showed better protection than VP1 and nFliC". 
One of the main problems is that the authors failed to evaluate the protective efficacy of DHAV-3 VP1 in duckling, properly. First, the number of molecules in 80 µg of tagged VP1-C protein is not the same number of molecules in 80 µg of tagged VP1 protein. According to the manuscript (page 4 lane 155), the predicted molecular weights of tagged VP1-C and tagged VP1 are 46 kDa and 28 kDa, respectively (which is doubtful, see below comments). Therefore, around 1.6 times more molecules of antigen were inoculated to the tagged VP1-C groups, compared with tagged VP1groups. It should be better to use the number of moles (mol) for the vaccination experiments to compare the immunogenicity of the antigens with different molecular weights. Second, all the recombinant proteins used in this study were fused with 6xHis- and S-tags as well as TrxA-tag. N-terminal of the proteins was TrxA-, 6xHis- and S-tagged and their C-terminal was 6xHis-tagged. Therefore, it is obvious that anti-TrxA tag, anti-6xHis tag and anti-S tag antibodies were also produced in the vaccinated ducklings. In addition, the vaccine formulations of VP1/VP1-C + nFliC contains double-amount of TrxA-6xHis- and S-tag regions compared with the formulations containing single protein. As the purified VP1 protein, which should be tagged protein, was used for ELISA to evaluate the level of antigen-specific antibodies, the all values shown in Fig. 2 are the sum of VP1-specific antibody and tag-specific antibodies. Thus, it is uncertain if inoculation of VP1-C + nFliC produced the highest level of VP1-specific antibody. Untagged VP1 proteins or DHAV particle (or lysate of DHAV-infected cells) should be used for ELISA to eliminate the reactions caused by tag-specific antibodies. In addition, these tags could affect cellular responses and cytokine elicitation. In particular, several thioredoxins are known to be involved in the immune responses. Therefore, the effects of tags must be excluded somehow in these experiments, too. For example, the proteins (which should be 20 kDa of N-terminal TrxA-6xHis-S-tagged enterokinase site) purified from the bacteria transformed with the empty pET32a(+) plasmids should be used as a control. Or, at least N-terminal TrxA-6xHis-tag region should be removed by thrombin-treatment prior to the vaccination experiments. 
The other problem is that the manuscript contains many incorrect information, which makes the data unreliable. For example, the primer set used to amplify the coding region of VP1-C (residues 149-218) is actually amplifying the coding region for a half of VP1 (residues 107-218). According to the SDS-PAGE analysis shown in Fig.1A, the migration rate of VP1-C (28 kDa) is slightly slower than nFliC (31 kDa). The reviewer understands that some proteins, in particular charged proteins, could migrate differently. However, the actual VP1-C is more likely the region between residues 107 and 218 (approximately 32.4 kDa). The region containing the predicted G-H loop (somewhere in the residues 130-150?), which is well known as the important antigenic loop. Therefore, the authors' claim in the discussion section (page 8 lanes 283-292) should be inappropriate. 
Likely, the authors purified the proteins only once. As the amounts of expressed proteins are easily change, it is hard to agree that the shorter VP1-C protein does allow higher expression amount (page 4 lanes 158-160 and elsewhere). The authors must properly analyze the expression levels of the proteins and show the data as mean ± SD. 
Discussion should provide an interpretation of the results in relation to previously published work and should not contain extensive repetition of the Results section or reiteration of the introduction. There are several reports for vaccine developments against DHAV infection. The author should emphasize the novelty and merits of their subunit vaccine.    
Further, much more data should have got from the sacrifice of 55 ducklings (11 ducklings x 5 groups). For example, the authors should show if the replication was not observed after DHAV challenge, if the vaccinated duckling showed any symptoms, how the antibody level changed after challenge and so on. 

-Minor comments-
There are many minor errors, such as typos, wrong usage of references, wrong accession numbers, wrong labeling and so on. Please go through the whole manuscript carefully before submission.  

Author Response

(The authors gave the same response as above.)

Round 2

Reviewer 1 Report

Truong TN and Cheng LT provided data that duck hepatitis A virus genotype 3 (DHAV-3) VP1-C protein might be potential subunit vaccine. However, the data is not well present, and the authors should provide more data to support their conclusion. The quality of this manuscript is not good enough. Several specific comments are listed below.

  1. Even though the CPE was observed in DHAV-3 infected LMH cells, the authors need to characterize the viral antigen expression in virus-infected cells by immunofluorescent assay. The data is unsound.
  2. The antibody levels could be determined without quantitated positive control serum.
  3. The authors tried to clarify the meaning of individual letter in the revised manuscript. I suggested to use different symbols instead.

Reviewer 3 Report

The revised manuscript (vaccines-1455825) has been slightly improved, however, most of the reviewer's concerns are still not addressed. Therefore, unfortunately, the article is still not sufficient for Vaccines level.

  1. According to authors' responses, they have reconfirmed the primer sites for VP1-C and the sequence data of their expression plasmid for the both tagged VP1-C and concluded that the information written in the manuscript is correct, which is hard to agree (please see the attached PDF file and below 2 comments).

1) According to the manuscript (lanes 59-60, Fig. 1C), the amino acid (aa) sequence of VP1-C region defined by the authors is "GRLYIWT........KKWSRN". Therefore, the VP1 nucleotide (nt) sequence included in the forward primer for VP1-C region must encode the first 4 or 5 aa residues (GRLYI) of this region. However, in fact, according to Table 1, the VP1 nt sequence of VP1-C forward primer, "actgctatggggggtatt" is translated to aa residues "TAMGGI", which obviously matches with the aa residues in the region 107-112 of VP1. The reviewer wonders which expression plasmid was used in this study.   

2) According to the Table 1, a total of 4 primers for VP1 and VP1-C were designed based on complete sequence (Accession number KU860089.1) of Duck hepatitis A virus (DHAV) 3 strain NC. However, nt sequences of 3 primers mismatch with KU860089.1 sequence. Each of VP1-C forward primer and VP1 reverse primer contains 1 nt mismatch (c to t and g to a, respectively) and VP1-C reverse primer contains 2 nt mismatches (tt to cc), which results in aa change from K to R at the residue 214. As 214th (in fact 215th) aa residue of VP1 of DHAV 3 strain NT01, which was newly isolated in this study, the reviewer wonders if the primers were designed based on the nt sequence of NT01. If so, Table 1 should be amended. In addition, the VP1 sequence of NT01 strain must be deposited in GenBank.

  1. According to authors' responses, the ducklings were inoculated with a total of 80 µg recombinant proteins. The vaccine formulations of VP1/VP1-C + nFliC contains 40 µg of each protein, which must be mentioned in the manuscript. It is hard to realize that the amount of each protein is 40 µg as there is no mention about it.

  1. As the reviewer mentioned in previous comment, it is uncertain if inoculation of VP1-C + nFliC produced the highest level of VP1-specific antibody. The reviewer agrees that all 4 vaccine formulations contain tags. However, the amounts (mol) and the structures of tagged recombinant protein in these formulations are obviously different. As far as the reviewer knows, larger amounts of antibodies against tags are usually produced in vaccinated animals. However, if the tag regions in the tagged recombinant proteins are structurally difficult to be recognized by immune system (for example, tags are covered with the other parts of proteins), less amounts of antibodies against tags could be produced. Therefore, it is hard to agree with that the difference among antibody levels of the vaccine groups should still be attributable to the DHAV-derived regions of the recombinant proteins unless the antibody titers for tags are precisely evaluated. The phrase "VP1-specific antibody" in the manuscript should be considered. The virus neutralization assay would be the easiest way to prove the highest level of VP1-specific antibody production in VP1-C + nFliC inoculated group.

  1. According to the manuscript and authors' responses, the authors speculate that the full-length tagged VP1 protein could be misfolded during refolding step. The reviewer wonders if the antibody titers can be precisely determined by ELISA using misfolding tagged VP1 protein?

  1. The aa positions are required in Fig. 1C.

  1. According to the authors' responses, the pathology from infection was recoded and confirmed. However, the reviewer could not find any pathological description in the result section. To evaluate the protective efficacy of vaccine, in particular subunit vaccine using outer layer of the pathogen, the neutralization antibody production and the virus replication as well as clinical signs should be first monitored, which is standard tactics for vaccine development. In addition, even ducklings vaccinated with VP1-C + nFliC were slowly dying (25% in 15 days). Therefore, if the virus was still growing in the vaccinated ducklings, it may be possible that the inoculation of VP1-C + nFliC just gives infected ducklings a few weeks' grace of death or allow them to go around and spread viruses?

  1. Please confirm that the statistical analysis was performed properly. It is slightly difficult to agree that there is no significant difference in antibody production PBS inoculated group and other group at day 7 post vaccination (Fig. 2), in IFN gamma and IL-6 expression between VP1 and VP1+ nFliC inoculated groups and in IFN gamma and IL-12 expression between PBS and VP1 inoculated groups.

  1. Since this study focused on vaccine development, it would be better to construct phylogenic tree of DHAV VP1 protein (aa sequence) rather than nt sequence. In addition, the maximum-likelihold method would be appropriate rather than neighbor-joining method?

  1. According to the aa sequence of NT01 shown in Fig. 1C, the region in the VP-C (184PMRSNEL190) is absolutely different from other related strains (for examples, NC and YT1213: 183HTLLNKI189, 090818: 183HTLLNKV189). The reviewer wonders if the inoculation of VP1/VP1-C of NT01 together with nFliC can protect the ducklings from the infection with other DHAV3 strains?

  1. There are still many minor errors found in this manuscript.
